# Hot Lithography Vat Photopolymerisation 3D Printing: Vat Temperature vs. Mixture Design

**DOI:** 10.3390/polym14152988

**Published:** 2022-07-23

**Authors:** Farzaneh Sameni, Basar Ozkan, Hanifeh Zarezadeh, Sarah Karmel, Daniel S. Engstrøm, Ehsan Sabet

**Affiliations:** 1Wolfson School of Mechanical, Electrical and Manufacturing Engineering, Loughborough University, Loughborough LE11 3TU, UK; f.sameni@lboro.ac.uk (F.S.); b.ozkan@lboro.ac.uk (B.O.); d.engstrom@lboro.ac.uk (D.S.E.); 2Additive Manufacturing Centre of Excellence Ltd., Derby DE23 8YH, UK; 3Photocentric Ltd., Peterborough PE1 5YW, UK; hanifeh@photocentric.co.uk (H.Z.); sikarmel@gmail.com (S.K.)

**Keywords:** vat photopolymerisation, hot lithography, polymerisation shrinkage, degree of conversion, glass transition temperature

## Abstract

In the vat photopolymerisation 3D printing technique, the properties of the printed parts are highly dependent on the degree of conversion of the monomers. The mechanisms and advantages of vat photopolymerisation at elevated temperatures, or so called “hot lithography”, were investigated in this paper. Two types of photoresins, commercially used as highly accurate castable resins, with different structural and diluent monomers, were employed in this study. Samples were printed at 25 °C, 40 °C, and 55 °C. The results show that hot lithography can significantly enhance the mechanical and dimensional properties of the printed parts and is more effective when there is a diluent with a network T_g_ close to the print temperature. When processed at 55 °C, Mixture A, which contains a diluent with a network T_g_ = 53 °C, was more readily impacted by heat compared to Mixture B, whose diluent had a network T_g_ = 105. As a result, a higher degree of conversion, followed by an increased T_g_ of the diluents, and improvements in the tensile strength and dimensional stability of the printed parts were observed, which enhanced the outcomes of the prints for the intended application in investment casting of complex components used in the aero and energy sectors. In conclusion, the effectiveness of the hot lithography process is contained by a correlation between the process temperature and the characteristics of the monomers in the mixture.

## 1. Introduction

Additive manufacturing (AM), otherwise known as 3D printing, is a manufacturing technique in which 3D parts are produced directly from a CAD model in a layer-by-layer fashion [1]. Stereolithography or vat photopolymerisation is one of the most commonly used, and was the first commercialised AM method in which a photosensitive resin (photoresin) was used as the feedstock material loaded in a tray, which is often referred to as the “vat”. In bottom-up vat photopolymerisation 3D printing, the vat has a transparent film on its bottom plate that is located on a screen from which a light source (usually between 355–470 nm) illuminates the photoresin and cures it layer-by-layer to manufacture the desired 3D part [2,3,4,5]. Liquid crystal display (LCD) 3D printing, also known as masked stereolithography (MSLA), is a lithography-based AM technique in which LCD screens are used to cure photoresins in a layer-wise fashion [6]. The type of photoinitiator used in the liquid photoresin system depends on the wavelength of the light source used. Photoinitiators can be UV sensitive (<400 nm) or visible-light sensitive (400 to 790 nm). The efficiency of the polymerisation process depends on the structure and reactivity of the monomers and the photoinitiators [1,7]. (Meth)acrylate monomers, which are commonly used in vat photopolymerisation systems are polymerised through free radical polymerisation [8]. Photoinitiators in a photoresin absorb the photolytic energy and produce reactive species or free radicals. These reactive species initiate the polymerisation process by also converting the monomers into free radical species. As the polymerisation process continues, more monomers are added to the polymer chain, which is known as the propagation step. Finally, the polymerisation process terminates in three ways: two polymer chains combine and produce one large chain; radicals cancelling each other or disproportionation; and the entrapment of radical ends within a solid polymer network due to the reduced mobility [8,9]. There is a wide range of monomers with (meth)acrylate functional groups that can be used in photocurable systems. In order for a crosslinked network to be created, the monomers should be multifunctional with at least two reactive (meth)acrylate groups [10]. In general, (meth)acrylate-based photoresins used in 3D printing have a high polymerisation rate when paired with suitable photoinitiators and exposed to light with a wavelength that corresponds to the photoinitiators [11]. Different types of (meth)acrylate monomers and oligomers can be introduced to a mixture to adjust the thermal and mechanical properties of the final polymer [12,13,14]. However, one of the drawbacks of using (meth)acrylate monomers is their high shrinkage, which results in warping and curling defects [9]. This limitation is more evident when insufficient polymerisation occurs due to reaching the gel point at low conversions (at around 20%), which slows down the polymerisation to the point that further polymerisation is prevented. This early gelation leads to a diffusion limitation, thus restricting the free flow of monomers and active species, and hence vitrification occurs in low conversions [9]. Vitrification is the stage after gelation in which the increased cross-linking density and polymer chain length results in the formation of a glassy solid and the polymerisation slows down considerably [15]. Delaying the gelation and vitrification to higher conversions can significantly reduce the shrinkage-related residual stresses and reduce the deformation of the printed objects [9,11,15]. 

The (meth)acrylate viscosity is a function of their molecular weight and structure. Viscosity of the photoresin mixture influences its degree of monomer conversion, as lower viscosity delays the gelation [10,16,17]. As the viscosity of the monomers increases, their mobility decreases, and therefore lower monomer conversion occurs before the gelation stage. Although this suggests that the lower molecular weight monomers offer higher conversion rates, monomers and oligomers with higher molecular weights have superior mechanical properties, even at lower conversion, hence being referred to as “structural monomers” [3,4,9]. The presence of structural monomers often leads to higher viscosities, and they need to be diluted with low molecular weight monomers for a balanced viscosity, strength, and conversion rate. The viscosity of the photoresin is even more important when it is used as a base media to load particles (known as filled resins). The lower viscosity of the resins enables the higher amounts of filler loading where necessary [18,19].

Applying heat to photopolymer systems has proven to affect the polymerisation rate due to the reduced viscosity [1,11,13,17,20]. Hot lithography, in which the lithography process is conducted at elevated temperatures, has introduced opportunities to enhance the mechanical properties of the parts by increasing the monomers’ mobility and enhancing the conversion rate of the process. The effect of hot lithography on the mechanical properties of the printed parts has been extensively studied, and the use of custom-oligomers with superior mechanical properties has been implemented in studies by several researchers [18,21,22] and commercialized by CubiCure GmbH. However, one of the main limitations of photopolymer-based AM methods is the shrinkage and warpage of parts due to the use of (meth)acrylate monomers. In this study, we carried out a systematic study on the effect of heat on the shrinkage of the 3D printed parts and suggesting ways to overcome this limitation.

In this paper, we investigated the impact of heat on two different photoresin mixtures that are commonly used in photopolymer formulations for the 3D printing of investment casting patterns, which require high dimensional accuracy and stability. The viscosity of the two mixtures was studied under different shear rates and temperatures. Samples were prepared at three process temperatures of 25 °C, 40 °C, and 55 °C to study: (1) the degree of conversion; (2) the tensile properties; (3) the glass transition temperature; and (4) the dimensional behaviour including polymerisation shrinkage, susceptibility to deformation, and dimensional stability. Therefore, the result of this study helps scientists and practitioners to tailor the formulation of their base photoresins for enhanced mechanical and dimensional properties and reduced shrinkage using hot lithography vat photopolymerisation 3D printing.

## 2. Materials and Methods

### 2.1. Materials and Mixture Preparation

Two photoresin mixtures (Mixture A and Mixture B), each comprising a structural monomer and a reactive diluent monomer, were designed for this study. The weight percentage of the structural to diluent components in both mixtures was kept at 70:30, respectively. To make the mixtures photocurable, 1 wt.% (on top of the total monomer weight) of phenylbis (2,4,6-trimethylbenzoyl) phosphine oxide (BAPO) (Rahn AG, Zürich, Switzerland) was added to each mixture as the photoinitiator. A total of 1.5 wt.% (on top of the total monomer weight) of Photocentric Black Pigment Stock (Photocentric Ltd., Peterborough, UK) was added to both mixtures to control the light penetration and therefore the curing properties of the photoresins.

Mixture A contains a difunctional urethane dimethacrylate (UDMA) (Rahn AG, Zürich, Switzerland) as the structural monomer. Triethyleneglycol dimethacrylate (TEGDMA) (BASF plc, Stockport, UK) was incorporated in this formulation as the reactive diluent due to its low viscosity.

Mixture B was formulated by using the trifunctional Tris(2-hydroxyrthyl) isocyanurate triacrylate (THEICTA) (IGM Resins, Wynyard, UK) as the structural component and dipropylene glycol diacrylate (DPGDA), (BASF plc, Stockport, UK) as the reactive diluent. To achieve a homogenous distribution of all components, the mixtures were mixed using a planetary high-speed mixer (SpeedMixer DAC 150 FVZ, Hauschild, Hamm, Germany) for 15 min at 20,000 rpm. Table 1 presents the properties of the monomers used in this study, the molecular structures are shown in Figure 1, and Table 2 summarises the formulations of the mixtures.

### 2.2. Hot Lithography 3D Printing

An in-house manufactured temperature control system was assembled on a Photocentric Liquid Crystal Precision 1.5 3D printer, which had a wavelength of 440 nm and a light intensity of 1.2–1.3 mW cm^−2^. Briefly, three electric heating fans (12 v) were installed on the 3D printer’s cover. The temperature controller system (Inkbird ITC-308) controlled the function (on/off) of the fans through a contact thermometer probe, which was placed inside the vat and measured the temperature of the photoresin constantly and maintained it at the set point (25 °C, 40 °C, and 55 °C in this case). Figure 2 shows a schematic of the hot lithography setup in this work. For all test specimens, the layer thickness was set to 100 µm, and printing temperatures were set to 25, 40, and 55 °C, respectively. The viscosity of the mixture at these temperatures was measured as discussed in the next section, followed by the degree of conversion measurements as well as mechanical and dimensional behaviour of the parts printed at these temperatures. Test parts were investigated in their green and post-cured states. Post-curing was achieved by heat treating the samples in an oven at 80 °C for 180 min. 

### 2.3. Viscosity Measurements

The viscosity measurements were taken using a TA AR-1000 (TA Instruments, New Castle, DE, USA) rheometer. Tests were conducted by loading less than 2.0 mL of the sample onto the temperature-controlled static holding plate with a 100 µm gap size to the opposite stainless-steel parallel plate with a diameter of 40 mm. The viscosity was measured using two methods: (1) Under a constant shear rate of 50 s^−1^ and a temperature range of 25 to 60 °C, and (2) under a variable shear rate between 0–50 s^−1^.

### 2.4. Depth of Cure Measurements

In order to measure the curing properties of the mixtures in relation to temperature, the method introduced by Steyrer et al. [18] was employed. In this method, each photoresin mixture within the vat was heated to the test temperature and then illuminated by the LCD screen in the shape of a 10 mm diameter disk. Since the light intensity of the LCD screen in this study was not amendable, the exposure time was changed between 10, 15, and 20 s at each temperature. The thicknesses of the cured disks were then measured using an external micrometer (Mitutoyo Corp, Kawasaki, Japan). The tests were conducted in triplicate and the mean values were used (Table 3).

### 2.5. Degree of Conversion

The degree of conversion (DC) of the photoresins was measured using the attenuated total reflection (ATR) module of a FTIR spectrometer (Bruker Alpha, Platinum ATR, Billerica, MA, USA) with 40 scans and a resolution of 2 cm^−1^. The uncured photoresins were used as the reference and the cured disks that were made in the previous section for the depths of the cure measurements were used as the test samples. The DC was calculated by Equation (1), in which A_C=C_ is the peak area at 1635 cm^−1^ for C=C double bonds, and A_C=O_ is the peak area at 1720 cm^−1^ as the internal standard for the DC calculations in both the uncured photoresin (M) and cured polymers (P).
(1)$$\mathrm{DC}=1-\left[\frac{{\left(\frac{A_{C=C}}{A_{C=O}}\right)}_{P}}{{\left(\frac{A_{C=C}}{A_{C=O}}\right)}_{M}}\right]\times 100$$

### 2.6. Tensile Properties

The effect of the process temperature on the tensile properties of the parts printed using Mixture A and Mixture B was studied in both the green (G) and post cured (PC) states, and in the X and Z printing directions. Samples were prepared according to the specifications of ASTM D638. Five tensile specimens of type V were prepared for each set. Tests were conducted on an Instron 6800 universal testing machine (Instron, Norwood, MA, USA) with a test speed of 5 mm/min. 

### 2.7. Dynamic Mechanical Analysis (DMA)

DMA was used to measure the glass transition temperature, using the Tan δ peak, of the printed test parts in the XY direction. Samples of all three temperatures were tested in their green state and one additional post-cured specimen for each mixture was tested as a reference. Tests were carried out on a TA Analysis Q800 DMA machine from room (current) temperature to 200 °C for Mixture A and 250°C for Mixture B, with a heating rate of 3 °C/min, a frequency of 1 Hz, amplitude of 20 µm, and an initial force of 0.1 N.

### 2.8. Dimensional Studies

The dimensional accuracy and stability of the parts printed at different temperatures were studied using a test model similar to the one introduced by Weng et al. [23] (Figure 3). Having two length levels (30 and 50 mm) in the X and Y directions, this test part allowed for measurements in two sizes in each direction at once. Four test parts were printed and measured for each print temperature. Three types of measurements are reported to represent:
Polymerisation shrinkage: For this purpose, the dimension test parts [23] were measured on the platform. The absolute dimensions of the as-printed parts on the platform are reported as an indication of their polymerisation shrinkage.Susceptibility to deformation: For this purpose, the same samples used in the previous step were measured before and after detachment from the build-platform while in the green state. The deviation (in percentage) is taken as an indication of the “accumulated internal residual stresses”, which are partially released upon removal from the platform [24] and therefore their susceptibility to deformation as a result of “stress relaxation” (in the green state).Dimensional stability: Same parts were aged in an oven and away from light at 25 °C for 7 days and measured again. The difference between the dimensions of the parts before and after the 7-day ageing period was taken as an indication of their dimensional stability (reported in percentage).


## 3. Results and Discussions

### 3.1. Depth of Cure Measurements

As other works [18,22] have indicated, at elevated temperatures, the reactivity of a given photoresin system increases and therefore higher curing depths can be achieved for a constant exposure time at elevated temperatures. The addition of black pigment stock to the photoresins used in this study resulted in a reduced light penetration and depth of cure of the photoresins and, therefore, the increased temperature had a less pronounced influence on their curing properties. Hence, as shown in Table 3, increasing the temperature from 25 °C to 55 °C with an exposure time of 10 s yielded a depth of cure between 230 and 260 µm for Mixture A, which is not considered as a significant improvement. For Mixture B, however, increasing the temperature from 25 °C to 55 °C resulted in a 5 s decrease in exposure time, from 15 s to 10 s, in order to obtain a cure depth of 250 µm. As suggested by [18], the printing layer thickness was taken as half this cure depth to ensure a fully solidified polymer. The depth of cure of a set of blank control samples for each mixture (without any black pigment addition) was also measured. These measurements highlight the effectiveness of the addition of black pigment in controlling the depth of cure of the samples in the presence of heat. Table 3 presents the measured depth of cure of Mixture A and Mixture B at their corresponding process temperatures; measurements were taken in triplicate and average values are reported.

### 3.2. Degree of Conversion

Disks with a thickness of 250 µm were used to measure the DC of the cured photoresins at different process temperatures. As summarised in Table 3, the disks reached 250 µm of cure depth in 10 s of exposure time for Mixture A at all three temperatures, while for Mixture B, a 15 s exposure was employed for 25 °C and 40 °C, and 10 s for 55 °C to reach the same cure thickness of 250 µm. As shown in Table 4, the DC of both mixtures cured at 25 °C and 40 °C showed minimum improvement with heat and were very close. However, increasing the process temperature from 40 °C to 55 °C introduced a significant increase of 55% to the DC of Mixture A. Unlike Mixture A, curing Mixture B at 55 °C had a small impact on its DC. The ATR spectra of the uncured photoresins and their networks at different temperatures are presented in Figure 4. The different responses of Mixture A and Mixture B to heat can be associated with their molecular structures as well as their network T_g_ values, as shown in Table 1. As suggested by Yu et al. [25], when the vitrification occurs in the glass forming polymer networks, the free volume within the network freezes and reduces the mobility of the reactive species, and therefore polymerisation comes to an end. Section 3.5. expands on the correlation between the monomers’ T_g_ and their response to heat. 

### 3.3. Viscosity Measurements

Figure 5 and Figure 6 present the results of the viscosity measurements of the photoresins. As expected, by increasing the temperature from 25 °C to 60 °C, the viscosity of the mixtures decreased significantly. It should be noted that more significant viscosity reduction (more than 300 mPa.s) was achieved when the temperature increased from 25 °C to 40 °C compared to when it further increased from 40 °C to 55 °C (a change of less than 100 mPa.s in viscosity) (Figure 5), whilst the main impact of hot lithography, as discussed in the previous section, and in Section 3.4 and Section 3.5, was in the higher temperature range (from 40 °C to 55 °C). Furthermore, despite sharing a similar range of viscosity at 55 °C (Figure 6), Mixtures A and B showed significantly different DCs at this temperature. These indicate that the impact of reduced viscosity on the effectiveness of hot lithography is contained and cannot be the main reason for the improved properties. This will be further discussed in Section 3.5.

### 3.4. Mechanical Properties

The impact of the process temperature on the ultimate tensile strength (UTS) of the test specimens is shown in Figure 6. In general, the mechanical properties of the printed parts are expected to increase with an increase in DC [26]. For Mixture A, the mechanical strength of the samples after post-curing showed only a little dependency on the process temperature in the X-axis, while a 19% improvement was observed in the samples printed in the Z-axis. However, when it came to the green strength in both the X and Z axes, significant improvements were observed when the process temperature was elevated. The UTS of the Mixture A-55 (Mixture A printed at 55 °C) test specimens in the X-axis improved by 28% compared to Mixture A-25 in their green state. This figure was significantly higher for the Z-axis printed test specimens, with their UTS doubling when printed at 55 °C, as shown in Figure 7a.

While several researchers have reported the improved mechanical properties of vat photopolymerisation parts as a result of elevated temperatures [15,18,22], its substantial impact on the green strength of the parts in the Z-axis is a significant finding of this study and is of great importance in practice. The green strength is the mechanical strength (Young’s modulus) of a 3D printed part during the printing process and is therefore a crucial parameter for the success of the print. This outcome can be explained by the improved interlayer bonding between the individual layers [27]. In other words, printing at elevated temperature not only increases the cross-linking density at each layer (hence higher strength in the XY direction), but also improves the interlayer bonding (hence the higher strength in the Z direction).

Unlike the above discussion for Mixture A, the impact of the process temperature on the tensile strength of Mixture B showed a minimum improvement by inducing heat to the process, even at a higher range of 55 °C. This improvement was contained to a moderate rate of 15% in the XY direction and 17% in the Z direction, as opposed to 39% and 98% in the case of Mixture A in the XY and Z directions, respectively (Figure 7b). The underlying reason behind this observation is explained in the next section.

### 3.5. Dynamic Mechanical Analysis (DMA)

To better understand the effect of process temperature on the mechanical properties of the parts, DMA studies were conducted on the samples to measure their glass transition temperature (T_g_). The T_g_ of thermoset polymers is directly related to their DC and cross-linking density, and therefore higher T_g_ values are expected when photoresins are processed at elevated temperatures [25].

The Tan δ curves of the polymerised samples of both photoresins showed bimodal glass transitions (Figure 8). Tu et al. [13] attributed the bimodal glass transition to the vitrification of the trapped uncured resin volumes in the cross-linked network due to the heat applied during the analysis, rather than indicating the T_g_ of each component. In contrast, as explained by Stansbury [17], the structural diversity between the comprising monomers of a mixture results in polymerisation induced separation, which in turn leads to the formation of heterogenous polymer networks. Likewise, considering the temperature dependent behaviour in our T_g_ measurements, we also corresponded the bimodal glass transition to the T_g_ of the individual components of the photoresin mixtures. For this purpose, we refer to Figure 9 to explain each mixture in the context of its structural and diluent components.

Regarding Mixture A, the T_g_ of its cured structural component (UDMA, network T_g_ = 134 °C) at all process temperatures remained almost unchanged between 144 °C and 148 °C (Figure 9b). However, while the T_g_ of its diluent component (TEGDMA, network T_g_ = 53 °C) cured at 25 °C and 40 °C were almost the same and between 75 °C and 77 °C, curing at 55 °C considerably increased its T_g_ to 96°C (by 32%) (Figure 9a). Hence, the reduced viscosity of the monomers was not the main driver for this phenomenon, as the viscosity difference between 25 °C and 40 °C was much higher than that between 25 °C and 40 °C (Figure 5 and Figure 6a). Therefore, to capture the highest impact of the heat on photopolymerisation, the process temperature needs to be above a certain threshold, for which the network T_g_ of the monomers (at room temperature) can be used as a direct indicator. In this experiment, the process temperature of 40 °C did not show a significant impact on T_g_ of neither diluent nor the structural components, and consequently, the tensile strength of the parts, as shown in Figure 7 and Figure 9, respectively. However, increasing the temperature to 55 °C (closer to the network T_g_ of the diluent) introduced a substantial increase to the diluent’s T_g_, and hence the tensile strength of the printed parts. Therefore, the improved UTS of the parts printed at 55 °C can be induced by a higher cross-linking density of this mixture caused by TEGDMA, the diluent component that has a lower T_g_ value.

To further validate this observation, Mixture B was also processed at 25 °C, 40 °C, and 55 °C, respectively.) As the reactive diluent of Mixture B (DPGDA) has a higher original T_g_ value of 102 °C (Table 1), the T_g_ of this component showed a minor increase of 12% when the process temperature was raised from 25 °C to 55 °C (Figure 9a). Similar to Mixture A, the T_g_ of the structural monomer of Mixture B (THEICTA, Network T_g_ = 240) remained unchanged at all three process temperatures (Figure 9b). Once again, these figures confirm that in order to have an effective hot lithography, the mixture must contain at least one component with a network T_g_ close to the process temperature. In both Mixture A and Mixture B, the obtained T_g_ values were in excellent agreement with the results of our DC measurements and the tensile tests.

To sum up, when choosing the process temperature, aside from the reduced viscosity, the proximity of the process temperature to the network T_g_ of the monomers inside the photoresin during printing is essential. In this case, applying heat will assist the unreacted radicals to further reach out for other counterparts, and thus form longer polymer chains and denser networks [28]. Table 5 summarises the T_g_ measurements of Mixtures A and B, which were associated with their structural and diluent components.

### 3.6. Dimensional Studies

The dimensional accuracy and stability of a cross-shaped specimen were studied at two lengths and three different stages: (1) Parts on the build platform; (2) parts detached from the build platform in the green and post-cured state; and (3) after ageing for 7 days in a dark place at 25 °C. Similar to previous sections, Mixture A was processed at three temperatures: 25 °C, 40 °C, and 55 °C and Mixture B at two temperatures: 25 °C and 55 °C.

Polymerisation shrinkage

The measured lengths of the as-printed parts before detachment from the platform for each set of experiments are shown in Figure 10. For both mixtures, increasing the process temperature resulted in higher shrinkage in the parts, which can be explained by the increased monomer conversion and cross-linking density [28] polymerisation [9]. Similar results were achieved in the work presented by Jongsma et al. [16], in which the volumetric shrinkage increased as the polymerisation temperature increased. This shrinkage, which is linear and two dimensional for each newly cured layer, can be compensated for by scaling the 3D model.

2.Susceptibility to deformation

Unlike the similar trend of polymerisation shrinkage between Mixture A and Mixture B, the dimensional changes of the parts when removed from the build platform showed different trends (Figure 11). Regarding Mixture A, which proved to be more readily affected by the temperature range used in this study, increasing the process temperature to 55 °C generally resulted in less dimensional change in the parts when they were detached from the build platform (Figure 11a). This can be explained by the increased T_g_ and the strength of the parts printed at this temperature (due to higher DC), and hence the increased resistance to deformations [22]. However, when printed at a mid-range temperature (40 °C), the parts showed a different trend, and their dimensional changes were similar to, or higher than, those printed at 25 °C (Figure 11a). Although compared to 55 °C, the samples printed at 40 °C had less polymerisation shrinkage (and therefore less stress accumulation) as they had a lower DC, and their T_g_ and UTS had not been enhanced to the level of the parts printed at 55 °C, and were instead closer to those of the parts printed at 25 °C. As the rate of stress formation when the conversion increases is higher than the rate of T_g_ development with increased conversion [15], samples printed at 40 °C showed mechanical properties and a T_g_ closer to those printed at 25 °C, while their stress accumulation due to the polymerisation shrinkage was higher, therefore, more deformation was evident compared to the samples printed at 25 °C and 55 °C.

As polymerisation stress is higher when the monomer conversion happens under constrained conditions [17] (this is induced by the previous layer in vat photopolymerisation), delaying the gel point and especially the vitrification, this allows for higher free monomer conversion before a rigid and glassy-like polymer network is formed [17] in polymerisation. Consequently, as a result of a less constrained conversion at elevated temperature, less polymerisation stress is built up in the parts printed by hot lithography [17,28].

For Mixture B, based on the results of the DC and T_g_ measurements, dimensional studies were only conducted at 25 °C and 55 °C, where the impact of heat was more obvious. The samples showed higher deformation when processed at 55 °C compared to 25 °C (Figure 11b) such as printing Mixture A at 40 °C. This is justifiable with a similar reasoning: where the T_g_ and strength of the parts has not adequately improved, constraint polymerisation shrinkage causes a higher level of residual stress in the parts.

3.Dimensional stability

In both mixtures, printing at elevated temperatures resulted in less dimensional change after ageing for 7 days, therefore, more stability (Figure 12). The highest dimensional stability was associated with Mixture A samples with the highest DC, which were printed at 55°C. Higher conversions during the printing process benefits dimensional stability of the parts in two ways: (1) Higher T_g_ and modulus was achieved, therefore the parts were stiffer and more resistant to dimensional change after printing; (2) achieving higher conversions during the printing process reduces the “dark polymerisation” after printing and causes less polymerisation while ageing, therefore improving the dimensional stability of the parts. Dark polymerisation is defined as the continued polymerisation after the light source is removed [29]. Once again, Mixture A showed more dependency on the heat, and the impact of the process temperature on the stability of the parts was more pronounced compared to Mixture B.

The results of the dimensional studies in the present study are consistent with the outcomes of the DC measurements, mechanical testing, and T_g_ measurements, and highlight the advantages of employing heat in the photopolymerisation process where high dimensional stability is desired.

## 4. Conclusions

This paper studied the effects of elevated temperature on vat photopolymerisation, or so-called hot lithography 3D printing. Increasing the process temperature significantly reduced the viscosity of the photopolymer mixtures. Lower viscosity improved the printability and reduced the separation forces during the printing process. The degree of conversion, tensile strength, glass transition temperature, and dimensional behaviour of two commercially available (meth)acrylate photoresin mixtures used as castable resins for the 3D-printing of dimensionally accurate casting patterns (labelled as Mixture A and Mixture B) were investigated at room and elevated temperatures. The findings are:
Hot lithography can significantly enhance the mechanical and dimensional properties of the parts when the right print temperature is chosen for a given mixture. Monomers with a lower network T_g_ (closer to the process temperature) respond more readily to the applied heat during hot lithography. In this case, hot lithography leads to an enhanced DC, hence improving the mechanical properties of the mixture printed at that temperature. This is particularly useful as aside from reducing the mixture’s viscosity, which in turns lowers the separation forces, higher mechanical properties of the prints reduce the chance of deformation caused by the remainder of the separation and peeling forces during printing.Hot lithography at the right temperature can significantly reduce the susceptibility of the parts to deformation, which occurs after the printing process (e.g., when detached from the build platform), and later when aged or post-cured. This is particularly useful for applications in which the dimensional accuracy is of great importance.


Although this paper suggests that the lowest T_g_ in the mixture can be an indication for choosing the temperature of the hot lithography process, the T_g_ itself is a function of various characteristics of the monomer including (but not limited to) functionality, molecular structure, and molecular weight. This highlights the need for future work to further refine the effect of hot lithography on the factors influencing T_g_, with an aim to better understand the science of hot lithography.

## Figures and Tables

**Figure 1 polymers-14-02988-f001:**
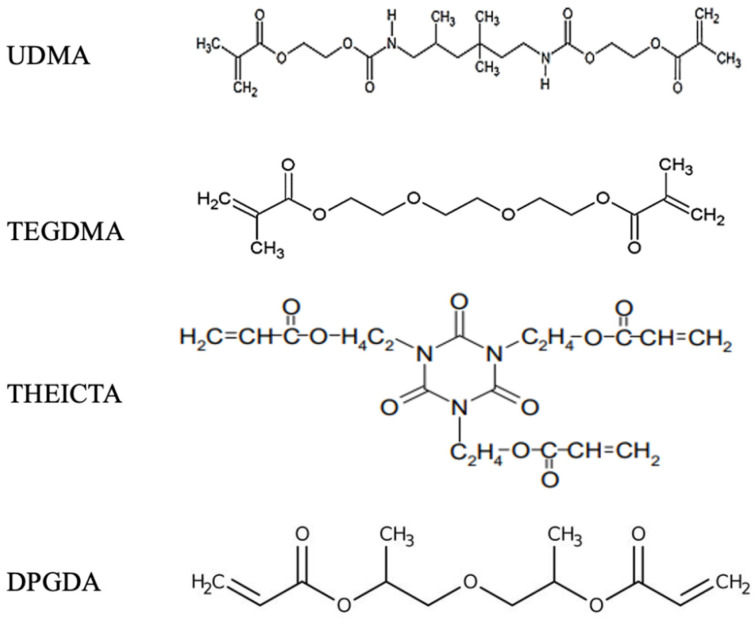
The chemical structures of the diluent and structural monomers.

**Figure 2 polymers-14-02988-f002:**
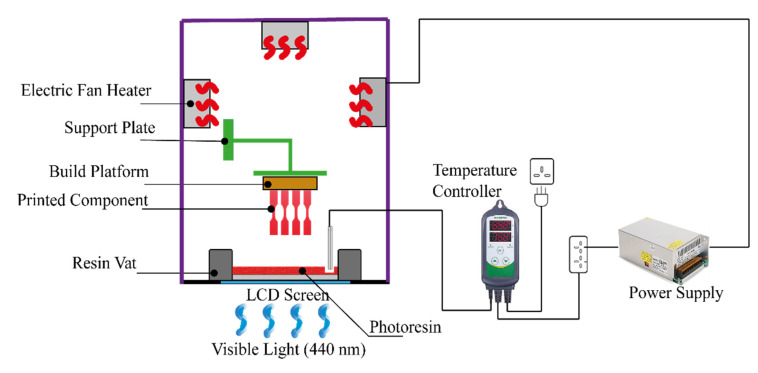
A schematic of the hot lithography setup used in this work.

**Figure 3 polymers-14-02988-f003:**
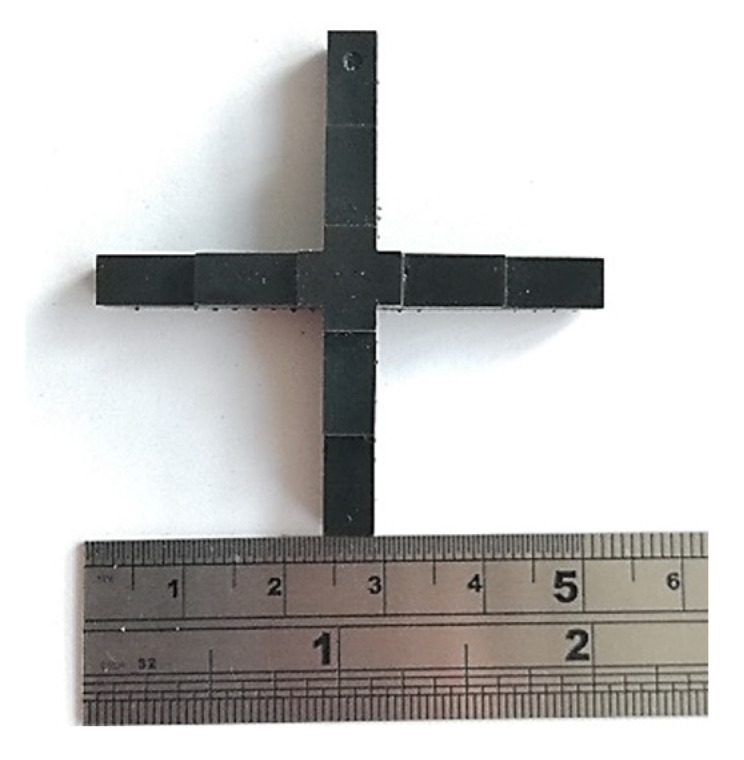
The printed cross-shaped dimension model used in this work, adapted from Weng et al. [23].

**Figure 4 polymers-14-02988-f004:**
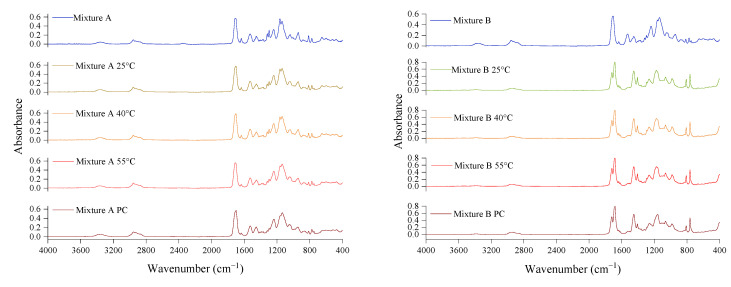
The ATR of the uncured photoresins (Mixture A and Mixture B) as well as their cured networks at different process temperatures.

**Figure 5 polymers-14-02988-f005:**
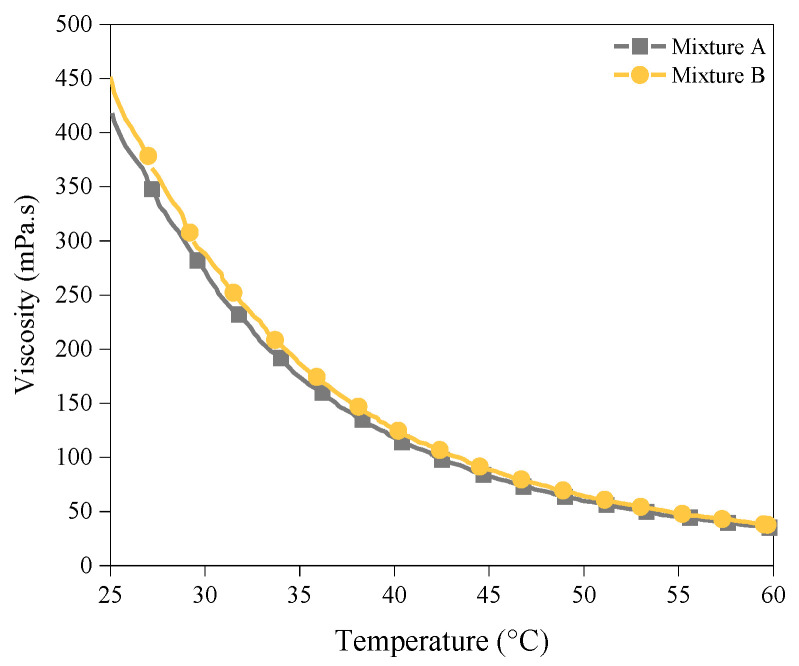
The changes in the viscosities of Mixture A and Mixture B by increasing the temperature from 25 to 60 °C under a constant shear rate of 50 S^−1^.

**Figure 6 polymers-14-02988-f006:**
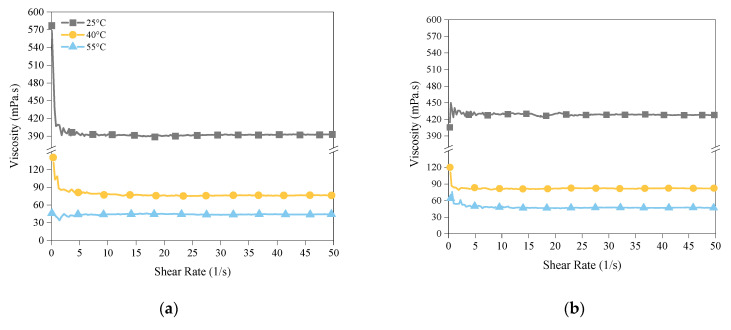
The viscosity of (**a**) Mixture A and (**b**) Mixture B at process temperatures of 25, 40, and 55 °C.

**Figure 7 polymers-14-02988-f007:**
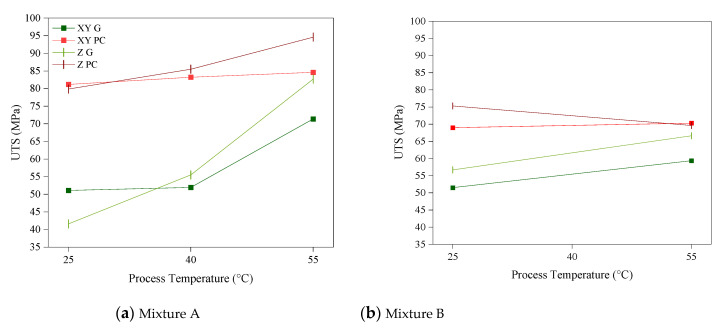
The effect of the process temperature on the UTS of (**a**) Mixture A and (**b**) Mixture B in the XY and Z directions, and the green (G) and post-cured (PC) states.

**Figure 8 polymers-14-02988-f008:**
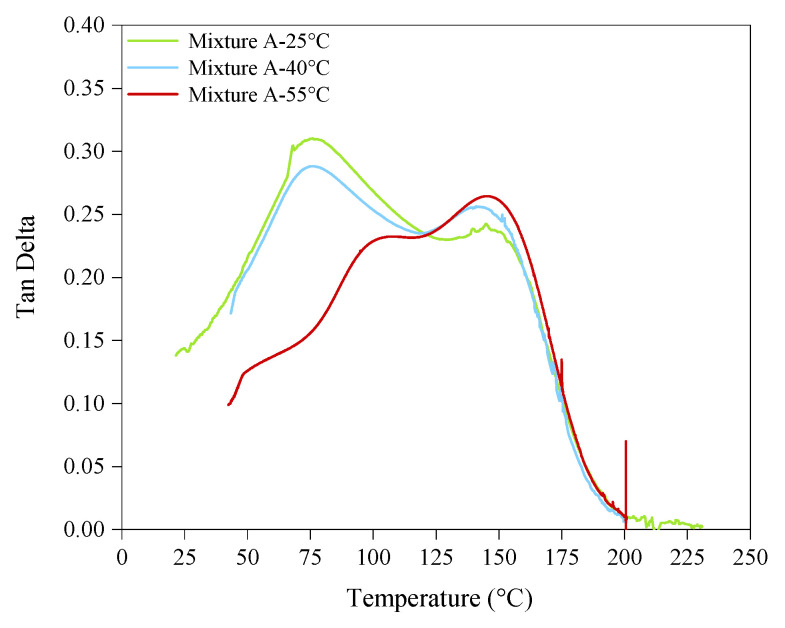
An example of the bimodal Tan delta curves obtained by DMA. The T_g_ of the mixtures were defined by the peak of tan δ curves.

**Figure 9 polymers-14-02988-f009:**
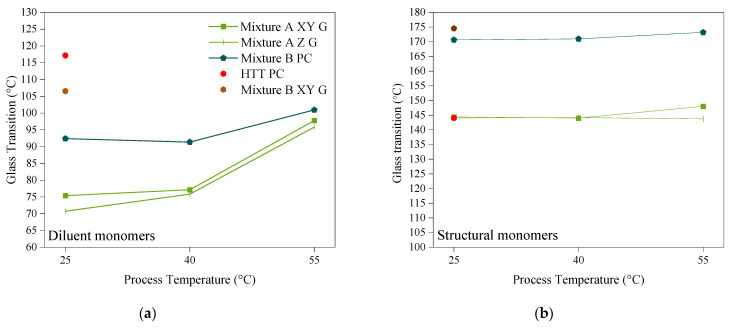
The changes in the glass transition temperature of the (**a**) diluent and (**b**) structural components of parts printed at 25, 40, and 55 °C using Mixture A and Mixture B. Samples were printed in the XY directions and tested in their green (G) state. One post-cured (PC) sample from each mixture was tested for reference.

**Figure 10 polymers-14-02988-f010:**
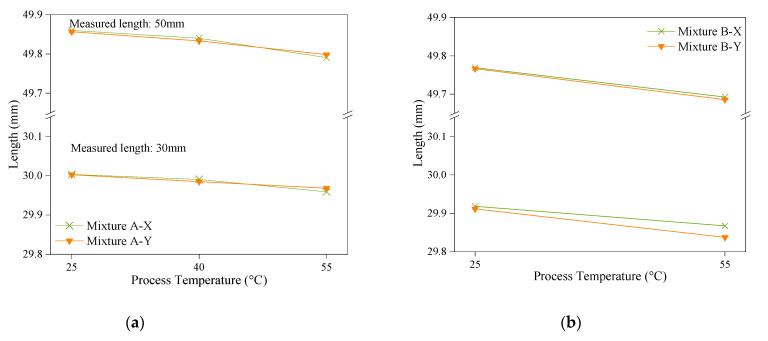
The impact of the process temperature on the measured lengths of the cross-shaped specimen at the 30 mm and 50 mm levels in the X and Y directions using (**a**) Mixture A and (**b**) Mixture B.

**Figure 11 polymers-14-02988-f011:**
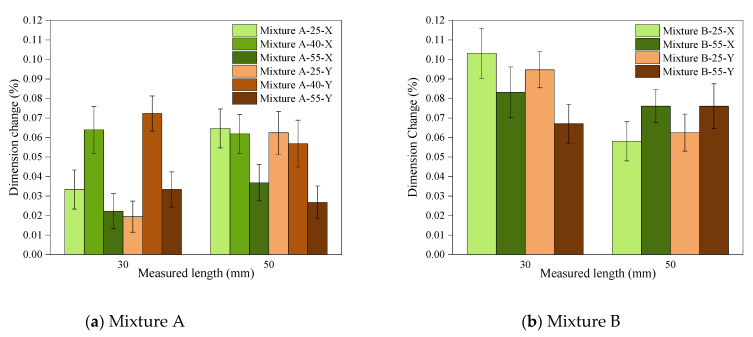
The dimension change of the cross-shaped model printed at different process temperatures upon removal from the platform (susceptibility to deformation), measured in the X and Y directions and at two lengths (30 and 50 mm).

**Figure 12 polymers-14-02988-f012:**
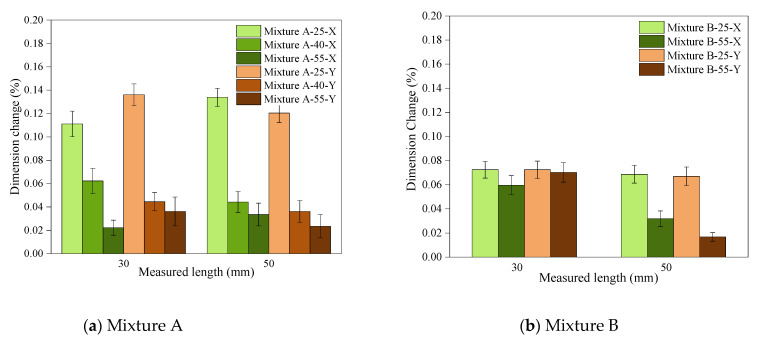
The dimension change of the cross-shaped parts after a 7-day ageing period as an indication of their dimensional stability.

**Table 1 polymers-14-02988-t001:** The characteristics and chemical structures of the monomers.

Monomer	Mw (g/mol)	Network T_g_ (°C)	Viscosity (mPa.s) *
Urethane dimethacrylate (UDMA)	470	134	10,000
Triethylene glycol dimethacrylate (TEGDMA)	286	53	5–30
Tris(2-hydroxyEthyl) isocyanurate triacrylate (THEICTA)	423	240	Crystalline solid
Dipropylene glycol diacrylate (DPGDA)	242	104	5–15

* Viscosity at 25 °C.

**Table 2 polymers-14-02988-t002:** The compositions of Mixtures A and B.

Monomer Content in the Mixtures (Total of 100 wt.%)					Other Additives on Top of 100 wt.% Monomer Mixture	
	TEGDMA (wt.%)	UDMA (wt.%)	DPGDA (wt.%)	THEICTA (wt.%)	BAPO *	Black Pigment Stock *
					(wt.% on Top of the Monomer Mixtures)	
Mixture A	30	70	-	-	1	1.5
Mixture B	-	-	30	70	1	1.5

* Photoinitiator and black pigment were calculated at 1 wt.% and 1.5 wt.% in addition to the total monomers weight, respectively.

**Table 3 polymers-14-02988-t003:** The depth of the cure measurement (µm) of Mixtures A and B at different temperatures.

Exposure Time (s)		Mixture A (µm)			Mixture B (µm)		
		25 (°C)	40 (°C)	55 (°C)	25 (°C)	40 (°C)	55 (°C)
	3 *	270	480	520	252	461	510
	10	230	250	260	200	220	250
	15	320	370	370	240	250	300
	20	410	420	440	300	330	390

* Control samples for each mixture. (no black pigment)

**Table 4 polymers-14-02988-t004:** The degree of conversion of Mixtures A and B at different process temperatures and after post-curing (as reference) calculated using the ATR spectrum.

	Degree of Conversion (%) at Process Temperature			
	25 °C	40 °C	55 °C	25 °C, Post-Cured
Mixture A	36.01	36.45	56.78	72.95
Mixture B	33.13	36.65	37.18	56.12

**Table 5 polymers-14-02988-t005:** The T_g_ associated with the structural and diluent components of the two photoresins at different process temperatures and in their post-cured state.

Process Temperature (°C)	Glass Transition Temperature (°C) of Photoresin Components			
	Mixture A		Mixture B	
	TEGDMA	UDMA	DPGDA	THEICTA
25	75.39	144.41	92.38	170.65
40	77.16	144	91.35	171
55	97.78	148.42	100.94	173.21
Post-cured	106.55	144.01	117.17	174.53

## Data Availability

Data are available on request due to restrictions. The data presented in this study are available on request from the corresponding author.

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
