# Peer review of "Hot Lithography Vat Photopolymerisation 3D Printing: Vat Temperature vs. Mixture Design"

_polymers, 2022, doi:10.3390/polym14152988_

Round 1
Reviewer 1 Report
1) The concept, “the glass transition temperature (Tg) of monomer”, claimed in this paper is unreasonable. Regarding to the monomer, including the 4 types of (meth)acrylate as structural and diluent functions, has no glass transition phenomena in the temperature up and down process. Only amorphous polymers have the glass transition behavior. Therefore, explanations of flexibility of molecular structure and lower viscosity before curing, from section 3.2, 3.4 and 3.5 and 3.6, is not convincing. Authors need to demonstrate/support the evaluated temperature leads to higher monomer mobility and polymerization rate with more appropriate reasoning.
2) In section 3.1, authors indicate that the black pigment stock was applied to control the light penetration depth. Please add a blank control, Mixture A and B without adding black pigment, to show the evaluated temperature effects on the depth of curing. Also, from Table 3, in Mixture B’s result, the increase of temperature from 25 °C to 55 °C leads to the depth of curing increasing about 30 – 40 % regardless of exposure time, which disagrees with the statement in line 178 and the result of Mixture A. Please add more explanation.
3) The degree of curing is crucial in this study. Please attach the ATR-FTIR diagrams to support the results in section 3.2.
4) What is the curing condition of photopolymerization during hot lithography 3D printing? Is there any inert gas (Nitrogen) protection to reduce the oxygen inhibition? The oxygen inhibition effect may result in depth of curing, degree of curing and the mechanical properties.
5) In this paper, the Mixture A and Mixture B are two types of molecules system, one is methacrylate, and the other one is acrylate. What is the reason the authors did not use the same functional group system to study the effect of different structures? Regarding methacrylate and acrylate monomers, the authors may consider the difference in photopolymerization or curing kinetic under 25 °C, 40 °C and 55 °C when making comparisons between these two systems. It is suggested that the author should add discussion about the reaction kinetic difference between two types of monomers, so the effectiveness of evaluated temperature in photoresins structure is more pronounced.
Author Response
On behalf of all authors of this manuscript, I’d like to sincerely thank you for your thoughtful review and comments you made on our manuscript. We find all the comments fair and immensely helpful to make this manuscript significantly better, both in presentation/formatting and its technical and academic content. We hope the revised manuscript meets your expectations for a publishable article in this journal. Below, please find our replies to your comments.
1) The concept, “the glass transition temperature (Tg) of monomer”, claimed in this paper is unreasonable.
Thanks for your comment. We also believe that in this context Tg has no meaningful concept for the monomers and it only applies to their cured networks, and this is what the text intended to convey, but we appreciate that the terminology used in original manuscript caused confusion and, therefore, it has been now corrected throughput the revised manuscript.
2) Regarding to the monomer, including the 4 types of (meth)acrylate as structural and diluent functions, has no glass transition phenomena in the temperature up and down process. Only amorphous polymers have the glass transition behaviour. Therefore, explanations of flexibility of molecular structure and lower viscosity before curing, from section 3.2, 3.4 and 3.5 and 3.6, is not convincing. Authors need to demonstrate/support the evaluated temperature leads to higher monomer mobility and polymerization rate with more appropriate reasoning.
We appreciate the point made by the reviewer. The effect of temperature on liquid (meth)acrylate mixtures has been widely studied and the supporting literature references have been added to the revised manuscript. [15, 16, 18, 25, 58]. These references and our supporting arguments have been now added to the revised manuscript. Please see pages 7, 10, 11. To add to what has been revised in the paper, we wish to point out that the effect of reduced viscosity as a result of heat on the improved polymerisation of photocurable mixtures is widely studied by other researchers. However, in this work we demonstrated that two different monomer systems which have very similar viscosities at elevated temperatures (e.g., 55°C), do not show similar photocuring behaviours; hence reducing viscosity by increasing the process temperature cannot be the only factor in photopolymerisation. This observation was followed by a clear correlation between the Tg of the individual monomers (when cured) and the effectiveness of the applied heat. However, despite introducing the Tg of the (cured) monomers as a primary indication of the effectiveness of the hot lithography, we acknowledged that Tg itself is a function of various factors. Each of these factors could well be the substantial reason for the different responses of monomer mixtures to heat. Therefore, a separate study is currently being carried out by the authors to understand which of these factors are the key players in the effectiveness of hot lithography, and the underlying mechanisms. However, we decided to publish our current results at this stage to initiate a line of academic discussion, as there is only a handful of research papers on hot lithography despite its importance on dimensional accuracy and stability, and mechanical properties of the print objects.
3) In section 3.1, the authors indicate that the black pigment stock was applied to control the light penetration depth. Please add a blank control, Mixture A and B without adding black pigment, to show the evaluated temperature effects on the depth of curing. Also, from Table 3, in Mixture B’s result, the increase of temperature from 25 °C to 55 °C leads to the depth of curing increasing about 30 – 40 % regardless of exposure time, which disagrees with the statement in line 178 and the result of Mixture A. Please add more explanation.
Thank you for your comment and suggestion of adding measurements of the control samples.
Rightly pointed out by the reviewer, looking at the cure depth measurements of Mixture B in different temperatures shows a maximum of 30% increase (25°C vs 55°C, at 20s exposure time). Comparing this with the newly added measurements of the control samples in the revised manuscript, it highlights the effectiveness of the black pigment addition and offers more clarification.
With only 3 seconds of exposure time, the impact of heat on the cure depth of the control samples is massively intensified by 83% from 25°C to 40°C, and by 102% from 25°C to 55°C.
The corresponding data and discussion have been now added to table 3 and section 3.1 of the manuscript.
4) The degree of curing is crucial in this study. Please attach the ATR-FTIR diagrams to support the results in section 3.2.
The ATR diagrams of the samples have been now provided in figure 4 in the revised manuscript.
5) What is the curing condition of photopolymerization during hot lithography 3D printing? Is there any inert gas (Nitrogen) protection to reduce the oxygen inhibition? The oxygen inhibition effect may result in depth of curing, degree of curing and the mechanical properties.
We agree with the reviewer that oxygen inhibition can drastically affect polymerisation. However, the vat photopolymerisation 3D printer machine used in this study is a bottom-up system in which the curing process of each layer happens at the bottom of the tank (tray of material or so-called vat). This has the advantage of protecting each new layer from the direct contact with the atmospheric oxygen. Considering this, the experiments in this study were conducted under temperature-controlled ambient conditions. The cure test samples were also prepared in a similar condition at the bottom of the vat as mentioned in section 2.5 of the manuscript. The aim of this study is to mimic a ‘real world’ environment in a 3D printer. Currently, all commercial photopolymer 3D printers work under atmospheric conditions, i.e. with air rather than an inert gas and atmospheric pressure. But a few of the existing commercial printers have the option of temperature control, which we have also used in our study.
6) In this paper, the Mixture A and Mixture B are two types of molecules system, one is methacrylate, and the other one is acrylate. What is the reason the authors did not use the same functional group system to study the effect of different structures? Regarding methacrylate and acrylate monomers, the authors may consider the difference in photopolymerization or curing kinetic under 25 °C, 40 °C and 55 °C when making comparisons between these two systems. It is suggested that the author should add discussion about the reaction kinetic difference between two types of monomers, so the effectiveness of evaluated temperature in photoresins structure is more pronounced.
We wish to thank the reviewer for this very valid point.
The main objective of this work was to study the impact of hot lithography on the mechanical and dimensional properties of two main commercial photopolymers available in the market as solutions for castable resins, hence the formulations have been fixed in the paper to evaluate these solutions. This has been now added to the revised manuscript (Page 2) for further clarification. One of the mixtures has (Meth)acrylate structural and acrylate diluent, while the other one has acrylate structural and (meth)acrylate diluent. But regardless of the mixture content, the interesting observation of this study was the correction between the process temperature and the lowest Tg of the cured monomers of the mixtures, as explained in the paper. This paper has now led us to conduct further research on the science of hot lithography, in which we are designing a set of experiments to fully comprehend the effect of different factors such as chemical structure, functional groups (acrylate or methacrylate), functionalities, network Tg, and molecular weight of the monomers, in a systemic study. The result of this study will later follow this manuscript to further shed light on the effects and mechanisms of hot lithography vat-photopolymerisation. However, we believe this paper paves the way for the next manuscript, and is highly relevant to the special issue of your esteemed journal on Polymers for AM.
Reviewer 2 Report
The paper presents detailed results on the effects of vat temperature (in the range 25 - 55°C) during 3D printing on the mechanical properties, dimensional stability, degree of conversion, and various other factors related to the polymer physical-chemistry of two photo-curable resin compositions. The two mixtures, results for which are presented, were based on pairs of structural and reactive monomers, differing in their own glass-transition temperature, which was used as the main descriptor to explain the main findings reported in the paper, namely - that photo-curable resin compositions, based on a diluents with a lower Tg (triethylene-glycol dimethacrylate, in this case) are affected to a higher extend by the vat temperature, and in hand may exhibit more desirable mechanical properties and dimensional stability, when processed at the highest temperature (55°C).
On the positive side - the authors have demonstrated academic rigour and have in-depth results on the effects of vat temperature for both studied mixtures and have accumulated results from every possible side and angle, that may have an effect on the resin behaviour during 3D printing, and in hand - on the final print. The data in the manuscript surely will be of interest to the readers of Polymers, since it intertwines polymer chemistry and materials science with a novel topic, such as photolithographic 3D printing, which finds not only important applications not only in industry, but even from a hobby / makers perspective.
However, apart from the above-mentioned praise, I cannot recommend the manuscript for publication, given its current state, unless it’s greatly improved. While the results are there (and to some extent their interpretation) - the manuscript is a bit tedious to read. The text is bloated with brief, general statements - e.g. starting with the Introduction “(Meth)acrylate-based photoresins have a high polymerization rate when exposed to light …”, and then going throughout the Results and discussion section, which is a bit discussion heavy (which has some advantages, but in this case dilutes the focus on the results themselves). And ultimately - for me the biggest drawback is that some important information is lacking (e.g., no mention on the mechanism of photo-polymerisation are included, nor are details about the illumination wavelength and specifics included /* the 3D printer is treated as a black box in this case */, there is also little information about how heating is achieved and the experimental setup, etc.).
Unfortunately, I cannot bring more concise recommendations for improving the above mentioned deficiencies, so I have listed a few recommendations (on the cosmetic side) below:
(1) The abstract could benefit from a bit of shortening and clarity improvements, especially the text between lines 17-25 which is difficult to follow.
(2) “Mechanical properties in AM processes” does not seem to be very clear as a keyword.
(3) In order to guide unfamiliar readers, the authors, can introduce where the “vat” comes in “vat photopolymerisation" prior its first use in line 44.
(4) The reference styling is a bit inconsistent, jumping from [X]-[Z], to [X],[Y],[Z], I’d guess that it is an effect from the reference manager being used and will be corrected in the final version.
(5) whom. et al is Pfaffinger (Line 74)
(6) Repetitive statements in lines 79-81 (“.. this paper presents …”, “In this paper, we investigate …”. Overall I think that a careful read-through of the Introduction section should be carried.
(7) The additives present in both mixtures (lines 98-101) could be shifted up a paragraph to the information about their general composition.
(8) What does the asterisk signify in Table 1, column 4 ?
(9) Table 2. is a neater way to present both mixtures compositions, however, the sum of all wt.% in it is >100.
(10) Steyrer et al should be ref. [22], not [1] in Section 2.4. Additionally, even if the intensity is not amendable, at least some information should be provided about the illumination wavelength and intensity (even according to the technical specification of the 3D printer).
(11) Equation numbering style is unusual, and inconsistent with MDPI style.
(12) In line 157 “Weng et al.” points to ref. [2], then, in Figure 1 it points to [21], however it is actually [26]. It seems that there is an issue with the reference numbering in a large part of the text, please check.
(13) There is a mismatch of the subsection numbering in the Results section. After 3.6, the next points is 4, up to 6, within the Results, followed by “4.Conclusions”, which in hand contains subsections/paragraphs numbered 7-9. Please correct.
Author Response
On behalf of all authors of this manuscript, I’d like to sincerely thank you for your thoughtful review and the comments you made on our manuscript. We find all the comments fair and immensely helpful to make this manuscript significantly better, both in presentation/formatting and its technical and academic content. We hope the revised manuscript meets your expectations for a publishable article in this journal. Below, please find our replies to your comments.
The paper presents detailed results on the effects of vat temperature (in the range 25 - 55°C) during 3D printing on the mechanical properties, dimensional stability, degree of conversion, and various other factors related to the polymer physical-chemistry of two photo-curable resin compositions. The two mixtures, results for which are presented, were based on pairs of structural and reactive monomers, differing in their own glass-transition temperature, which was used as the main descriptor to explain the main findings reported in the paper, namely - that photo-curable resin compositions, based on a diluents with a lower Tg (triethylene-glycol dimethacrylate, in this case) are affected to a higher extend by the vat temperature, and in hand may exhibit more desirable mechanical properties and dimensional stability, when processed at the highest temperature (55°C).
On the positive side - the authors have demonstrated academic rigour and have in-depth results on the effects of vat temperature for both studied mixtures and have accumulated results from every possible side and angle, that may have an effect on the resin behaviour during 3D printing, and in hand - on the final print. The data in the manuscript surely will be of interest to the readers of Polymers, since it intertwines polymer chemistry and materials science with a novel topic, such as photolithographic 3D printing, which finds not only important applications not only in industry, but even from a hobby / makers perspective.
We thank the reviewer for their great description and understanding of the paper, and the extent of the data collection and results that have gone into this manuscript.
However, apart from the above-mentioned praise, I cannot recommend the manuscript for publication, given its current state, unless it’s greatly improved. While the results are there (and to some extent their interpretation) - the manuscript is a bit tedious to read. The text is bloated with brief, general statements - e.g. starting with the Introduction “(Meth)acrylate-based photoresins have a high polymerization rate when exposed to light …”, and then going throughout the Results and discussion section, which is a bit discussion heavy (which has some advantages, but in this case dilutes the focus on the results themselves).
We acknowledge many unsupported and general claims/comments had been made in the first submission. We believe we have now addressed most/all of them in the revised manuscript with a careful review of the text and encourage the reviewer to very kindly reconsider their academic assessment of the manuscript.
And ultimately - for me the biggest drawback is that some important information is lacking (e.g., no mention on the mechanism of photo-polymerisation are included, nor are details about the illumination wavelength and specifics included /* the 3D printer is treated as a black box in this case */, there is also little information about how heating is achieved and the experimental setup, etc.).
Thanks for your comment. We can now confirm Photopolymerisation mechanism and more info about the 3D printer have been added to the revised manuscript – please see pages 2 and 4 respectively. A schematic of the 3D printer and the heating system is added to the manuscript as figure 2.
I have listed a few recommendations (on the cosmetic side) below:
(1) The abstract could benefit from a bit of shortening and clarity improvements, especially the text between lines 17-25 which is difficult to follow.
The abstract has been now edited to become shorter and clearer as suggested.
(2) “Mechanical properties in AM processes” does not seem to be very clear as a keyword.
The keywords have been amended in the revised manuscript.
(3) In order to guide unfamiliar readers, the authors, can introduce where the “vat” comes in “vat photopolymerisation" prior its first use in line 44.
It is now indicated in the text that the vat is the holding tank/tray where the photoresin as the process feedstock is loaded to (Page 1).
(4) The reference styling is a bit inconsistent, jumping from [X]-[Z], to [X],[Y],[Z], I’d guess that it is an effect from the reference manager being used and will be corrected in the final version.
The reference styling has now been corrected throughout the manuscript
(5) whom. et al is Pfaffinger (Line 74)
Thank you for rightly noticing the reference manager’s issues. All the comments regarding the reference manager issues including comments 5, 10, and 12 have been now resolved in the revised manuscripts.
(6) Repetitive statements in lines 79-81 (“.. this paper presents …”, “In this paper, we investigate …”. Overall I think that a careful read-through of the Introduction section should be carried.
Thank you for your comment, the introduction section and specifically the repetitive statements have been now revised.
(7) The additives present in both mixtures (lines 98-101) could be shifted up a paragraph to the information about their general composition.
The abovementioned paragraph has been moved higher in the revised manuscript for a better flow, as suggested. Please see page 3.
(8) What does the asterisk signify in Table 1, column 4 ?
This is to note the viscosity of monomers at 25°C. Clarifying note has been added underneath the table now.
In addition, table 1 has been now amended by taking the chemical structures out and assembling them in Figure 1, to make the table neater.
(9) Table 2. is a neater way to present both mixtures compositions, however, the sum of all wt.% in it is >100.
Thanks for pointing this out. We acknowledge that the table in its last format was confusing. However, as mentioned in the manuscript the PI and BP are added regarding the monomers weight. For more clarification, the manuscript has been now amended to add more clarity. Please see paragraph 1 page 3, and revised table 2.
(10) Steyrer et al should be ref. [22], not [1] in Section 2.4.
Additionally, even if the intensity is not amendable, at least some information should be provided about the illumination wavelength and intensity (even according to the technical specification of the 3D printer).
The wavelength and light intensity of the machine have been now added to the revised manuscript according to manufacturer’s information, please see page 4 section 2.2.
(11) Equation numbering style is unusual, and inconsistent with MDPI style.
Equation 1 has been now corrected to match the MDPI style.
(12) In line 157 “Weng et al.” points to ref. [2], then, in Figure 1 it points to [21], however it is actually [26]. It seems that there is an issue with the reference numbering in a large part of the text, please check.
The references have now been corrected throughout the manuscript.
(13) There is a mismatch of the subsection numbering in the Results section. After 3.6, the next points is 4, up to 6, within the Results, followed by “4.Conclusions”, which in hand contains subsections/paragraphs numbered 7-9. Please correct.
Thanks for this comment. This has been now addressed by amending the automatic numbering.
Reviewer 3 Report
The topic of the paper is interesting and the paper is generally well written. Some comments for the authors to consider when revising the paper.
-Include a picture and some description about the custom temperature control system. What type of heating element and shape is used? Where the heating element is located?
-L142, “Equation 1” can be placed to the right of the equation.
-The conclusion section can be shortened, and some contents might be moved to the discussion section.
-Include error bar in Fig. 8 and Fig. 9.
Author Response
On behalf of all authors of this manuscript, I’d like to sincerely thank you for your thoughtful review and the comments you made on our manuscript. We find all the comments fair and immensely helpful to make this manuscript significantly better, both in presentation/formatting and its technical and academic content.
The topic of the paper is interesting, and the paper is generally well written. Some comments for the authors to consider when revising the paper.
We wish to thank the reviewer for the positive feedback and their interest in our manuscript.
-Include a picture and some description about the custom temperature control system. What type of heating element and shape is used? Where is the heating element located?
Thank you for tour comment. A brief explanation of the heating and temperature control system has been now added to the section 2.2 of the manuscript. A schematic of the 3D printer, heating fans, and the temperature control system are depicted in the graphical abstract and the revised manuscript (please see figure 2).
-L142, “Equation 1” can be placed to the right of the equation.
This has been now addressed in the manuscript.
-The conclusion section can be shortened, and some contents might be moved to the discussion section.
Thank you for your comment, this has been now addressed in the revised manuscript.
-Include error bar in Fig. 8 and Fig. 9.
Figure 11 and figure 12 contain error bars now in the revised manuscripts.
Round 2
Reviewer 1 Report
According to the new version of the manuscript, I would like to accept this paper to be published. The author answered all questions and added the relative figures/contents. Overall, this manuscript is well written.
Reviewer 2 Report
I would like to thank & congratulate the authors for the patience and effort in implementing the recommendations, suggested by me and my fellow reviewers. I do believe that the presentation of the data within the manuscript is improved in the revised version and I can recommend it for publication. In the version I read there are still some minor issues for correction (e.g., the capital W in "Wt.%"), which will sure be corrected during copy-editing and final proof-reading. Keep on the good work.